# Evaluation of Linkers’ Influence on Peptide-Based Piezoelectric Biosensors’ Sensitivity to Aldehydes in the Gas Phase

**DOI:** 10.3390/ijms241310610

**Published:** 2023-06-25

**Authors:** Tomasz Wasilewski, Damian Neubauer, Marek Wojciechowski, Bartosz Szulczyński, Jacek Gębicki, Wojciech Kamysz

**Affiliations:** 1Department of Inorganic Chemistry, Faculty of Pharmacy, Medical University of Gdańsk, Hallera 107, 80-416 Gdańsk, Poland; 2Department of Pharmaceutical Technology and Biochemistry, Chemical Faculty, Gdańsk University of Technology, Gabriela Narutowicza 11/12, 80-233 Gdańsk, Poland; 3Department of Process Engineering and Chemical Technology, Chemical Faculty, Gdańsk University of Technology, Gabriela Narutowicza 11/12, 80-233 Gdańsk, Poland

**Keywords:** sensors, biosensors, QCM, peptides, aldehydes, linkers, spacers, bioelectronic nose

## Abstract

Recent findings qualified aldehydes as potential biomarkers for disease diagnosis. One of the possibilities is to use electrochemical biosensors in point-of-care (PoC), but these need further development to overcome some limitations. Currently, the primary goal is to enhance their metrological parameters in terms of sensitivity and selectivity. Previous findings indicate that peptide OBPP4 (KLLFDSLTDLKKKMSEC-NH_2_) is a promising candidate for further development of aldehyde-sensitive biosensors. To increase the affinity of a receptor layer to long-chain aldehydes, a structure stabilization of the peptide active site via the incorporation of different linkers was studied. Indeed, the incorporation of linkers improved sensitivity to and binding of aldehydes in comparison to that of the original peptide-based biosensor. The tendency to adopt disordered structures was diminished owing to the implementation of suitable linkers. Therefore, to improve the metrological characteristics of peptide-based piezoelectric biosensors, linkers were added at the C-terminus of OBPP4 peptide (KLLFDSLTDLKKKMSE-linker-C-NH_2_). Those linkers consist of proteinogenic amino acids from group one: glycine, L-proline, L-serine, and non proteinogenic amino acids from group two: β-alanine, 4-aminobutyric acid, and 6-aminohexanoic acid. Linkers were evaluated with in silico studies, followed by experimental verification. All studied linkers enhanced the detection of aldehydes in the gas phase. The highest difference in frequency (60 Hz, nonanal) was observed between original peptide-based biosensors and ones based on peptides modified with the GSGSGS linker. It allowed evaluation of the limit of detection for nonanal at the level of 2 ppm, which is nine times lower than that of the original peptide. The highest sensitivity values were also obtained for the GSGSGS linker: 0.3312, 0.4281, and 0.4676 Hz/ppm for pentanal, octanal, and nonanal, respectively. An order of magnitude increase in sensitivity was observed for the six linkers used. Generally, the linker’s rigidity and the number of amino acid residues are much more essential for biosensors’ metrological characteristics than the amino acid sequence itself. It was found that the longer the linkers, the better the effect on docking efficiency.

## 1. Introduction

Electrochemical methods have gained increasing attention in recent years as viable alternatives to traditional analytical techniques due to their inherent advantages like simple construction, low costs, ability to work online, short analysis time, and suitability for a versatile label-free analysis. The high number of deaths caused by multifactorial diseases (cancers, respiratory system diseases, cardiovascular disorders, infections, etc.) results mainly from late diagnosis, which effectively impedes treatment and significantly increases the cost of medical care [1,2]. Identification of the biomarkers, especially in exhaled air, has the potential to be applied in the diagnosis of many diseases, for instance, lungs, digestive system, oncological and systemic diseases [3]. The peptide-based biosensors gain significant interest in the diagnostics of diseases where selective and sensitive analysis of volatile organic compounds (VOCs) is necessary [4,5,6,7]. The investigations on biosensors for odorous substances focus on mimicking their biological counterparts in terms of sensitivity and specificity, which would enable the effective detection of selected biomarkers. Many natural olfactory systems offer a wide range of biological elements to take advantage of [8]. Moreover, there are numerous electrochemical biosensors utilizing peptides as the biorecognition layers for the detection of analytes, such as metal ions, proteins, nucleic acids, enzymes, etc. [9,10,11]. Difficulties in the implementation of natural olfactory systems for the construction of the biosensors result from their demanding production and poor stability [8]; hence, there are methodological and instrumentation approaches being elaborated, which are based on the application of a specific protein fragment of artificial olfactory receptors to design the biosensors. The QCM (Quartz Crystal Microbalance) electrodes with a suitably prepared surface can serve as a substrate for the preparation of SAMs (Self-Assembled Monolayers) [12,13]; with an efficient cleaning technique, a single biosensor can be used multiple times [14]. The utilization of small protein fragments in biosensors, such as ligand binding regions or synthetic peptides, is a new trend in the analysis of simple odorous compounds [15]. Odorant Binding Proteins (OBPs) are key elements of chemosensory systems, and they are excellent candidates to be applied in biosensors [16,17]. Detection of natural and synthetic volatile organic compounds (VOCs) with OBPs has great development potential; however, their application in bioelectronic systems is still in the initial stages [18]. Determination of the amino acid sequence and optimum length of peptide chain that mimics odorant binding sites provide effective binding of VOCs. Deposition of a peptide on a transducer enables effective docking of volatile ligands with high specificity and selectivity, e.g., aldehydes in the gas phase, providing new prospects for the development of biomimicking materials in the field of odor biosensors [19]. The biosensors with peptides in the receptor layer are considered as tools for effective, cheap, easy, and fast detection and/or monitoring of analytes in medical diagnostics.

Analysis of VOCs in exhaled air can provide fast, reproducible, and non-invasive diagnostics of many diseases via the identification of disease-specific changes in a sample profile and detection of volatile biomarkers at suitably low concentration levels [20]. Medium- and long-chain aldehydes belong to key biomarkers of lung cancer, which is the main cause of death in all regions of the world [21,22]. This type of cancer is characterized by increased activity of many proteins, which makes prognosis and diagnostics very difficult [23]. Relative concentrations of VOCs in breath are different between health and disease [21]. Detection of VOCs is a topic of interest in various disciplines, which resulted in the elaboration of numerous analytical approaches to precise VOC identification [24,25,26], including the samples originating from the patients [27,28]. Taking advantage of technological progress, it was shown that VOC profiles characterize both general, as well as pathological health states. Many classic techniques of VOC analysis are, in certain aspects, limited to research laboratories, but it is predicted that in the future generation of VOCs, profiles will be the basis of PoC tests and will allow early diagnosis of diseases based on analysis of exhaled breath. It should enable fast and effective decisions concerning treatment [29,30]. Progress in metabolomics allowed the identification of the key biomarkers responsible for respiratory tract diseases, including cancers [31,32]. Despite indisputable advantages of VOCs analysis techniques employed so far, including gas chromatography-mass spectrometry (GC-MS) and microchip-based methods [33,34], they are still burdened by inconveniences related to time- and labor-consuming sample preparation and analysis, which can additionally generate false negative and false positive results [35]. Therefore, it is important to elaborate on the methods for precise and reliable in situ screening tests for the identification of volatile aldehydes to make progress in PoC tests.

Characteristics of peptides and proteins are closely related to their three-dimensional structure. The conformations adopted by peptides with respect to ligands are determined by non-covalent intermolecular interactions, such as hydrogen bonds, van der Waals forces, π-stackings, as well as hydrophobic and electrostatic interactions. Modulation of peptide secondary structure can be achieved through modifications promoting the aforementioned interactions. Effective ligand binding requires a specific secondary structure. Thus, stabilization of the active folded form of peptides or proteins is important from the standpoint of maintenance and intensification of functions of these molecules depending on binding conditions [36]. The α-helix is the most common secondary structure element. Helixes play a pivotal role in the mediation of interactions between proteins. The helical structure is also typically found in OBPs and, therefore, present in OBPs-derived peptides, which are expected to bind particular ligands [37]. Moreover, the average length of helix domains in proteins is rather short and engulfs from two to three-helix turns (or from eight to twelve residues). These complexes suggest that the formation of short helixes is possible, which potentially take part in selective interactions with molecules. However, peptides rarely maintain their conformation after “extraction” from protein; a dominant part of their ability for specific docking of target ligands is lost because they adopt a set of own structures, not a native structure. Stabilization of the peptides in a helix structure is aimed not only at decreasing their conformal heterogeneity but also a significant improvement in their resistance to degradation. The regions stabilizing protein structure occur in different multi-domain protein complexes, the example of which is cellulosome [38]; however, the same complex contains unordered fragments (linkers) different with respect to amino acid sequences and lengths, from a few to several amino acid residues. Nevertheless, the reason for the variability is still not completely clear, but the length and amino acid sequence of those linkers can substantially affect enzymatic activity [39,40]. Special attention is paid to the investigation of different approaches to stabilization of α-helix conformation in peptides or to mimicking this domain using “scaffolds”, “linkers”, “spacers”, etc., [36,41,42]. The α-helix contains 3.6 residues per full turn, which results in the arrangement of the side chains in positions i, i + 3 and i, i + 4 on the same surface of the complex structure. The classic strategy of stabilization of α-helix conformation in peptides utilizes covalent bonds between these groups of the side chain. Cross-linking of the side chains employs, e.g., lactam, disulfide, and metallic bridges. The helixes with lactam and hydrocarbon connections are characterized by elastic transverse bonds. Entropy considerations suggest that rigid linkers can contribute to more stable helixes. It was found that a rigid aromatic linker, which fits the distance between side chains, provides much higher stability than an elastic linker [36]. It was also proven that the rigid linkers, which are shorter than the target helix step, led to more stable helixes [43]. It guides repeated evaluation of the influence of the linker’s length in the helixes cross-linked with the side chain [44]. Additionally, peptide stability can be optimized via peptide modification, such as cyclization or chemical modifications, including the application of D-amino acids (instead of proteinogenic amino acids), chemically modified amino acids [45], or other linking structures. Polypeptide chains bound by the cysteine thiol group do not possess such high surface packing density as for example ones containing butanethiol, but their characteristics resemble the systems existing in nature. Utilization of linkers was aimed at increasing stability of packing of peptide layers and thus at improving the specificity. The QCM data confirmed intermolecular interactions resulting in increasing viscoelasticity of the peptide SAMs [46]. The literature reports that the linkers composed of four amino acids, for instance 4 proline residues, can stabilize helix structure of peptides and increase their hydrophobicity. Consequently, introduction of those fragments to peptides can bring elevated packing of SAMs and improved specificity [38,41,47]. This article focuses on a comparison of the influence of different linker types on structure stabilization and packing density, thus on effectiveness of ligand vs. receptor docking. Two groups of linkers were investigated: (a) proteinogenic amino acids composed of glycine, proline and serine: GSGSGS, GGGGS, PPPP, PPP; (b) non-proteinogenic derivatives: 4 × βAla, GABA, 2 × GABA, 6-Ahx. Linkers used in this study are presented in the table below (Table 1).

Particular amino acid sequences, which are part of complex secondary and higher-order structures, usually take random, folded structures after isolation in a short peptide mode instead of the shape of their biological counterpart [46]. To cope with this problem, different techniques aimed at the maintenance and improvement of structural stability are applied. Stabilization of the peptides in a structure should not only decrease their conformational heterogeneity, but also significantly increase their resistance to degradation. What is interesting, many of the methods for enhancement of stability or activity of the peptides or their protection from the activity of enzymatic proteins used in supramolecular chemistry can also be found in nature, in the peptides of microorganisms origin, for instance, change into D- instead of L-amino acids, deamination, glycosylation, cyclization, N-formylation, N-acylation, amidation of C-terminal, removal/addition of amino acid residues [48]. The aim of the paper was an attempt to improve peptide-based QCM biosensors’ sensitivity to aldehydes in the gas phase via the addition of different types of linkers. Eight linkers with desired properties were selected from the tested group. The literature reports on the linkers consisting of amino acid residues, e.g., prolines, that stabilize the helix structure of peptides and increase their hydrophobicity, which increases the packing density of SAM and improves specificity with respect to ligands [41]. A scheme of localization of the aforementioned linkers in a structure of the peptide deposited on a secondary transducer is shown in the figure below (Figure 1).

Commercialization of peptide-based biosensing devices requires prior improvement of the mass-scale production methods and biosensors themselves to optimize metrological parameters in terms of selectivity and sensitivity. Computational modeling and structural analysis of the peptide sequences capable of binding particular VOCs play key roles. The utilization of short linkers makes it possible to significantly enhance selectivity/specificity and simultaneously minimize the problems related to cross-reactivity. Importantly, the orientation and conformation of peptides deposited onto a transductor are crucial for VOC detection, and those are influenced by the linker. Therefore, different types and lengths of the linkers can affect biosensor response due to altered properties of the recognition moiety [49]. According to our knowledge, this is the first study on the effect of linkers on peptide-based biosensors intended for VOC detection.

## 2. Results

### 2.1. In Silico Docking Simulations

The models of all sensors for in silico experiments were prepared using a two-step process. First, the peptide model was constructed using the PEP-FOLD server [50,51]. Next, it was combined with the specific linker and placed on the surface of the gold plate model. Multiple copies of the peptide were added to achieve the desired density. The system was initially equilibrated for 1 ns under NVT conditions and then under NPT conditions. Subsequently, it was subjected to a 100ns molecular dynamics simulation at 300 K using the GROMACS package and the GolP-CHARMM force field [52,53,54]. This force field is well-suited for replicating the process of peptide adsorption on the gold surface. The resulting MD trajectories were then analyzed, and the most representative structures of the sensor models were selected as receptors for subsequent docking calculations.

The models of the VOC molecules were initially constructed using the HyperChem software [55]. All docking simulations were then conducted using the Autodock 4.2 suite of programs, with the necessary parameter files, ligands, and receptors being prepared and processed using Autodock accompanying scripts [56,57]. The affinities of the VOCs to the specific sensor model were evaluated using a modified Autodock force field [58]. The size and position of the docking grid were adjusted to cover the central region of the receptor. For each ligand-receptor pair, 50 docking simulations were performed, and their results were clustered. The lowest energy clusters and the corresponding ligand poses were selected as the final results for analysis.

The comparison between in silico predictions and experimental results is presented in Figure 2. There is a consistent discrepancy observed, not only between experimental and calculated values but also there are some noticeable differences in the responses of physical sensors of some ligands in the group of peptides with proteogenic vs. non-proteogenic linkers. Namely, sensors with proteogenic linkers better differentiate between various ligands. Particularly TMA, p-anisaldehyde, p-tolualdehyde, and heptanal give significantly different responses in this group. On the other hand, for sensors with non-proteogenic linkers, the difference is less pronounced. In this case, most ligands, except for the aromatic ones (p-anisaldehyde and p-tolualdehyde), elicit similar responses. This suggests that despite the simplicity of receptors derived from short peptides, they are still capable, when combined with an appropriate linker, of exhibiting some specificity not only towards ligands belonging to some general class of molecules (such as aliphatic aldehydes) but also towards some specific structural features. This specificity diminishes when non-proteogenic linkers are used. Only ligands with different chemical characteristics (p-anisaldehyde and p-tolualdehyde) give significantly different responses in this case. The source of the specificity difference between sensors using proteogenic and non-proteogenic linkers is not clear. However, results from molecular dynamics simulations indicate that while OBPP4 peptides maintain their helical structures, when deposited onto the gold plate, these helices become tilted with respect to the surface normal. The extent of the tilt varies depending on the linker used. Peptides with proteogenic linkers exhibit a narrower range of tilt compared to the other group, potentially facilitating the assembly of receptors with more consistently structured binding sites and possessing some degree of specificity (Appendix A).

The discrepancy between experimental and calculated affinities can also be attributed to the distinct chemical reactivity of the ligands used. Since in silico methods, based on empirical potentials, do not take into account reactivities and only capture the first stage of ligand binding, namely the formation of a non-covalent ligand-receptor complex, when numerous specific interactions that are usually present in the real binding site are lacking, these methods tend to predict similar affinities for different ligands if they have similar sizes. However, in reality, the different reactivities may be responsible for significantly different responses noted from physical sensors.

### 2.2. Peptides Deposition

The degree of peptide deposition for each biosensor is presented in Figure 3. Calculations were performed with the Sauerbrey equation based on a difference in frequency before and after peptide deposition. As was shown in the previous article [59], the degree of deposition of biolayers on the transducer can affect receptor-ligand affinity, which was taken into account during the evaluation of selectivity towards VOCs.

### 2.3. Biosensors’ Responses to Aldehydes in Gas Phase

As was proved in the earlier studies [24], the biosensor with OBPP4 active element reveals high affinity to long-chain aldehydes, i.e., octanal, nonanal, and undecanal. The biosensors immobilized with selected peptides were tested with respect to a reference gas—nonanal. Kinetics of adsorption was analogous for all the biosensors exhibiting a rapid decrease in signal after injection of the gas, followed by a slower increase until a steady state. The change of QCM response in time is strictly related to the gas (sample and zero air) passing through the measurement chamber. The double plateau is associated with a stoppage of flow (elimination of advection). In contrast, overshoot after desorption is related to high-flow purging, which is used to shorten the regeneration time of the sensor. The individual stages of the sample analysis are shown schematically in Figure 4A, together with the working power of the pump passing the sample through the measuring system. The total measurement time depended on signal stabilization before and after injection, as well as on the desorption rate. Single measurement engulfed stabilization of a baseline, introduction of gas and adsorption, signal stabilization, desorption, and return to the initial conditions. Analytical data were characterized by a difference in frequency between the value recorded as the baseline (F_0_) prior to adsorption and the output frequency defined as F_R_, (ΔF = F_0_–F_R_). The figures presented below illustrate the response of biosensors with deposited peptides from two groups to volatile aldehydes with characteristic stabilization of plateau after establishing of equilibrium state.

Figure 4 and Figure 5 present exemplary responses of the biosensors to medium- and long-chain aldehydes. One can notice a plateau after a drop in frequency as a result of saturation of the biosensor’s chamber with a reference gas and then subsequently return to the baseline after purging with air. The most pronounced frequency changes were observed for the linkers from group I (Figure 4); for instance, the peptide with GSGSGS linker with respect to octanal, nonanal, and pentanal, revealed ΔF of 70, 55, and 68 Hz ± 5 Hz, respectively. At the same time, a difference in resonant frequency amplitude for the parent peptide (OBPP4) in the case of the aforementioned aldehydes reached 10, 10, and 9 Hz ± 2 Hz, respectively. In the case of the peptides from group II (Figure 4), the biggest change in frequency was observed for the peptide with linker 2 × GABA and was equal to 100 Hz for octanal, 130 Hz for pentanal and 120 Hz ± 12 Hz for nonanal. Moreover, biosensors based on peptides from group II revealed affinity to acetaldehyde, which was manifested by a change in frequency at the level of 30 Hz. Tested biosensors exhibited a selective response to aldehydes, especially to long-chain ones (octanal, nonanal, pentanal), which was confirmed by increased sensitivity to these compounds and lack of response to the remaining VOCs. Biosensors’ sensitivity to aldehydes and corresponding LOD values are shown in Table 2 and Figure 6. The responses to the remaining groups of compounds were observed only for higher concentrations of volatile substances above 300 ppm.

LOD [ppm] was estimated based on the calibration curve using Formulas (1) and (2):(1)$$LOD=\frac{3.3\cdot S_{Y}}{a}$$
(2)$$S_{Y}=\sqrt{\frac{{\left(Y_{P}-Y_{M}\right)}^{2}}{n-2}}$$
where: $S_{Y}$—standard deviation of the response [Hz], a—calibration curve slope [Hz/ppm], $Y_{P}$—value predicted using calibration curve [Hz], $Y_{M}$—measured value [Hz], n—the number of experiments The following parameters were determined for presented biosensors: sensitivity (S), the limit of detection (LOD), selectivity, determination coefficient calculated for the calibration curve, and linearity range. The first two parameters were substantially different in comparison to our previous study, where the OBP4-based biosensor was validated and further comprehensively determined in the previous study [60]. The lowest LOD was recorded for the biosensor based on peptide with GSGSGS linker detecting nonanal at 2 ppm ± 1 ppm, with a sensitivity of 0.4676 Hz/ppm. In the case of the compounds from the remaining chemical groups (trimethylamine, methanol, ethyl acetate), there was no increase in sensitivity as compared to the original peptide, or the response was observed only in high concentrations (Appendix A). Sensitivity and LOD for selected aldehydes were plotted against the number of atoms between the amino group and terminal carbonyl carbon atom of the linker (Figure 7).

## 3. Discussion

It is not completely clear how peptides behave in a gas environment; the structure and conformation of immobilized peptides, especially those mimicking larger protein structures (OBPs), are difficult to predict. That is why it is necessary to verify in silico results with experimental investigations and to confirm receptor vs. ligand affinity. The research was aimed at determination of the influence of the type and length of linkers on the affinity of the ligand to aldehydes and stabilization of the receptor part of a previously elaborated peptide (OBPP4) [60]. Peptide monolayers deposited on the gold surface exhibit certain similarities to alkanethiols [61]. While the structure of alkanethiol monolayers is mainly stabilized by covalent bonds, van der Waals forces, and hydrophobic interactions, the molecular interactions in peptide monolayers are more complicated due to the presence of different functional groups, including salt bridges, π interactions, hydrophobic and hydrophilic interactions, which allow stabilization of a structure [41]. Mutual inter- and intramolecular interactions open the possibility of a wide range of interactions, which are additionally stabilized by the inclusion of a selected group of linkers into the structure of the peptide deposited on a gold electrode of a transducer.

Elastic linkers consisting of proteinogenic amino acids are commonly utilized to construct multi-domain proteins and stabilization of molecules in solutions; however, their influence on peptide stability in the gas atmosphere has not been investigated so far [46,62]. The impact of length and character of the linker was assessed in silico and experimentally with respect to selected compound groups. Additional inclusion of proline residues provides hydrophobicity of the molecule. It was proved that the PPPP linker ensures close packing of the peptide chains on a gold surface owing to extended polyproline helix conformation. Molecular Docking (MD) simulations revealed that peptides with PPPP linker adopt an α-helix secondary structure in contrast to GGGG linker. Moreover, the GSGSGS linker was successfully applied in QCM biosensors and is believed to induce α-helix formation in peptide molecules [46,63]. Proteinogenic linkers can be therefore divided into rigid and helix-inducing (PPP, PPPP, and GSGSGS) and flexible with no impact on helix formation (GGGGS [64]). Non-proteinogenic amino acids are used to link, e.g., cytotoxic agents, radiometal chelators, biotin, fluorescent probes, and many more to peptides/proteins [65,66,67]. Linkers based on non-proteinogenic amino acids used in this study are known to be flexible owing to –CH_2_- groups being able to freely rotate around the C-C bonds [68]. Typical alkyl linkers used in QCM are lipoic acid and aliphatic carboxylic acids with a terminal mercapto group [69,70]. This study includes linkers with non-proteinogenic amino acids, namely 4 × βAla (βAla; H_2_N-(CH_2_)_2_-COOH), GABA (H_2_N-(CH_2_)_3_-COOH), 2 × GABA, and 6-Ahx (H_2_N-(CH_2_)_5_-COOH). Linkers differ in the number of methylene groups and, therefore, in length and flexibility. Peptide with glycine-rich linker (GGGG) was determined to be non-helical, and therefore it can be deduced that increasing linker flexibility (Gly < βAla < GABA < 6-Ahx) will not induce stability in peptide secondary structure. Peptide OBPP4 (KLLFDSLTDLKKKMSEC-NH_2_) was selected as a model compound based on our previous study on biosensors with HarmOBP7 fragments [19]. Among tested peptides, the OBPP4 was revealed to be the most selective to long-chain aldehydes (e.g., octanal) in gas phase. The aim of the study is to examine the effect of linker length and type on biosensors response to aldehydes, especially long-chain ones. Hypothetically, linker can affect peptide secondary structure and peptide distance from electrode surface. It is expected that peptides with linkers will be more exposed to molecules in analyzed gas samples and will be more likely to adopt secondary structure mimicking OBP.

The increased affinity of the peptide modified with linkers containing glycine and serine residues can be explained by its rigidity and stabilization of the peptide’s secondary structure. Based on the in silico results, it can be stated that the regions with amino acid linkers (GSGSGS, GGGGS, PPPP, PPP, 4 × βAla) probably form a helix, which stabilizes the receptor part of the peptides. Interestingly our in silico results for peptides modified with “GGGGS” and “4 × βAla” linkers are surprising. MD simulations performed by Nowinski A.K. et al. [41] revealed that the “GGGG” linker did not induce any specific secondary structure. Nevertheless, there are some crucial differences. Firstly, our linker contains additional serine residue, and secondly, Nowinski analyzed dilute peptides, whereas this study simulated Au-conjugated ones. Linkers “GGGGS” and “4 × βAla” are known for their flexibility; therefore, it is unexpected to observe helical structure. Probably there are interactions between peptides in a monolayer, and packing density can influence a conformation of the entire monolayer with a simultaneous impact on receptor vs. ligand affinity. It is expected that the presence of the molecules of volatile ligand (long-chain aldehyde) enforces the adoption of suitable conformation by a peptide, which is partially confirmed by the results of the biosensors’ response to VOCs. Removal of VOCs from the chamber by flushing with pure air is based on progressive desorption. Presumably, packing density can influence peptide conformation, and so the effectiveness of volatile ligands binding. Deposition levels of peptides with proteinogenic linkers were similar to that of original OBPP4 (Figure 3; 15.5–20.0 µg cm^−2^), while levels of that non-proteinogenic were ranged between 13.8 and 20.0 µg cm^−2^. The highest deposition was determined for 2 × GABA and the lowest for 4 × βAla. The literature review shows that the influence of packing density on docking affinity is a controversial subject [46], and some reports suggest a correlation between these two features [71,72]. Presumably, the highest packing density of OBPP4-2 × GABA effect in distinct affinity and, therefore, sensitivity to aldehydes. However, no general correlation between the deposition level itself and the metrological parameters of biosensors was observed. Interestingly, there is a linear correlation between the number of linkers’ back-bone atoms and biosensor sensitivity to aliphatic aldehydes (Figure 5), but this was only applicable to proteinogenic linkers. Similarly, the determined LOD of aliphatic aldehydes can be well-described by power laws functions where the function base is the number of linkers atoms. No such tendencies were observed for non-proteinogenic ones. To conclude, in the case of proteinogenic linkers, the LOD of aliphatic aldehydes will decrease, and sensitivity will rise with an elevating number of amino acid residues. Therefore, it can be presumed that OBPP4 with a longer peptide linker will be more effective in aldehyde-specific biosensors, e.g., GSGSGSG. Hypothetically, the GSGSGS linker, as it was argued in the literature [46], induces α-helix formation, which facilitates interactions with aldehydes. At the same time, it can be deduced that this linker provides the best peptide exposition to analyte molecules owing to the highest distance between the gold surface and the aldehyde binding site.

The incorporation of a 6-Ahx linker does not enhance the analytical performance of the biosensor in comparison to the original one. Surprisingly, the determined LOD for the biosensor with a 6-Ahx linker was distinctly higher in comparison to that of the OBPP4-based sensor, ranging from 200 to 300 ppm for long-chain aldehydes. Biosensors with linkers GGGGS, GABA, and 2 × GABA demonstrated slightly higher LOD (several ppm for octanal and nonanal) than biosensors with GSGSGS linker. The highest increase of LOD of octanal was observed for the PPP linker, where the difference between analogous LOD determined with OBPP4-PPP- and OBPP4-based biosensors is equal to 246 ppm. In contrast, the highest drop in LOD was observed for pentanal detection with the GSGSGS linker, where the LOD difference is 68 ppm. A decrease in LOD by 16 ppm ± 3 ppm with respect to the original peptide—OBPP4, means approaching the ppb order, which is a step towards the application of this type of biosensors to fast and effective detection of aldehydes as potential biomarkers of respiratory tract diseases. The LOD of studied biosensors (a few ppm) is comparable with the finest sensors from other studies (Table 3); however, the operation of chemoresistive sensors requires high temperature (200–300 °C) [73,74,75,76]. Some sensors, for example, the one designed by Tsujiguchi et al. [77], also revealed low LOD but required long exposure time to nonanal (5–24 h) and heating stage [74,75,76]. The closest metrological parameters are exhibited by the sensor designed by Jahangiri-Manesh et al. [78]. Utilization of molecularly imprinted polymers (MIPs) allowed the construction of a sensor with a LOD of 4.5 ppm for nonanal; the response was acquired in real-time without an initial stage of concentration and heating. According to one of the latest reports, the application of decorated monolayer WS_2_ [79] as the elements sensitive to nonanal molecules seems to be a promising solution. The next stage aimed at the evaluation of the usefulness of the designed sensor, combined into an array, will be an assessment of the response to real biological samples—exhaled air obtained from patients.

## 4. Materials and Methods

A piezoelectric transducer with a quartz plate (13.7 mm in diameter) and polished gold surface (5.1 mm in diameter) were used for all experiments. The AT-Cut 10 MHz QCMs with Au electrodes were acquired from OpenQCM (Novaetech s.r.l., Napoli, Italy). The frequencies before and after sensing were measured using OpenQCM Software. The resonant frequency of the bare QCM transducers was measured with QpenQCM Wi 2 (Novaetech s.r.l., Napoli, Italy) and a previously developed system [19]. All reagents and the volatiles (analytical grade) were purchased from Sigma-Aldrich (Sigma Aldrich Co., St. Louis, MO, USA). For purging, clean air was used (2 ± 1% relative humidity). Preparation of gas mixtures in Tedlar bags was carried out with zero air from GPZ-3B zero air generator (LAT Katowice, Poland). The air quality from this generator meets the following standards: EN 12619, EN 14211; EN 14212; EN 14625; EN 14626; EN 14662-3.

### 4.1. Peptides Design and Molecular Modelling

The OBPP4 peptide with KLLFDSLTDLKKKMSEC-NH_2_ sequence was employed in the investigations as a receptor element of the biosensor. It was selected based on previous in silico studies, which involved building a sensor model with the peptide mimicking aldehydes binding site in the HarmOBP7 protein present in the antennae of Helicoverpa armigera moth. Its affinity to aldehydes was experimentally proven [60]. A better reflection of the spatial structure of the peptide and mimicking of the native protein was attempted via various types of modification.

### 4.2. Peptide Synthesis and Deposition on QCM Transducers

All of the peptides were synthesized according to the established method [82] employing the solid-phase Fmoc/tBu strategy. Briefly, the synthesis was carried out on an automated microwave peptide synthesizer (Liberty Blue™, CEM Corporation, Mathews, NC, USA). The peptides were purified by reversed-phase high-performance liquid chromatography (RP-HPLC) with LP-chrome software. The original and purified peptides were analyzed by HPLC in a water/acetonitrile gradient. The purity (>95%) was confirmed by HPLC/UV-VIS (Varian, Mulgrave, VIC, Australia), and the identity by LC/MS (Waters Acquity SQD, Milford, MA). The lyophilized peptides were kept in the dark at 5 °C before deposition. The stability of biosensors’ layers was evaluated as previously described [19]. Biosensors between measurements were also stored in the same conditions.

When biomolecules such as peptides adsorb on the surface and thus increase the thickness, the device will receive a frequency change response [83]. According to the literature data, the degree of deposition can be evaluated based on changes in the resonant frequency before and after peptide immobilization, and it is expressed in ng·cm^−2^. Following the Sauerbrey Formula (3), it is possible to determine an exact change of peptide mass bound with the sensor [84]:(3)$$∆F=\frac{-2F_{o}^{2}∆M}{\left[{A\left(\mu_{q}\rho_{q}\right)}^{\frac{1}{2}}\right]}$$
where, $\rho_{q}$ and ${µ}_{q}$ are the density (2.648 g·cm^−3^) and shear modulus of quartz (2.947 × 10^11^ g·cm^−1^·s^2^), respectively, $F_{0}$ is the fundamental crystal frequency of the piezoelectric quartz crystal, A is the crystal piezoelectrically active geometrical area, which is defined by the area of the metallic film deposited on the crystal, ΔM and ΔF are the mass and frequency changes [83,84].

The purified peptide was deposited on the gold surface of QCM by a drop-casting technique. It is a reproducible, fast, and easily-accessible technique employing a relatively low volume of coating solutions and leading to a thin uniform layer [59]. The thickness of obtained coating depends on volume and degree of dispersion, concentration, properties of the solvent, and angle of contact between substrate and solvent. Then, the sensors were placed in a desiccator for 24 h. The peptides were immobilized due to the phenomenon of SAM formation on the gold surface. Organic compounds with thiol functional group (-SH) self-assemble on the surface of noble metals, decreasing free energy at the interface and forming densely-packed monolayers connected with gold by strong S-Au bonds [85]. A peptide solution was prepared using deionized water as a solvent. The peptide of concentration 0.5 mg·mL^−1^ and volume 20 µL was deposited on the gold electrodes’ surface of the QCM transducers (using an electronic pipette Eppendorf Xplorer (Eppendorf, Hamburg, Germany)). Each deposition process was carried out at room temperature following the method, which had been optimized before [59,60], which allows for reproducible deposition of peptides layers of piezoelectric transducers (Figure 3).

### 4.3. Measurement Setup

The biosensor with OBPP4 (peptide mimicking HarmOBP7 “binding pocket” region) receptor element was characterized in detail in the previous paper, where it revealed the lowest limit of LOD 14 ppm for nonanal [60]. VOCs at different concentration levels were prepared in Tedlar^®^ bags using a gas mixture generator [19]. The correctness of their preparation was verified with a gas chromatograph (430-GC, Bruker®, Bremen, Germany) according to the method elaborated earlier [19]. To evaluate the sensor’s sensitivity and selectivity to aldehydes, a series of experiments with formaldehyde, acetaldehyde, propanal, pentanal, hexanal, heptaldehyde, octanal, nonanal, decanal, undecanal; dialdehyde—glyoxal, benzaldehyde, p-tolualdehyde, p-anisaldehyde and 2-methyl,3-(3,4)-me- thylendioxyphenyl-propanal (helional)were performed. Based on a previous study [60], we focus on a group of aldehydes (pentanal, octanal, nonanal, acetaldehyde, p-anisaldehyde, heptaldehyde, p-tolualdehyde).

The measurements involved the BioEnos Q6 device (The Fahrenheit Union of Universities in Gdańsk—FarU), which measures and analyses frequency from six QCM transducers simultaneously. The transducers are placed in a measurement chamber made of polytetrafluoroethylene (PTFE), inside of which gas flow is enforced by a membrane pump localized before the outlet. The inlet of gases is controlled with a three-way valve from the sample line or flushing line equipped with a filter for the removal of potential air pollutants. Operation is governed by a microcontroller, which sets the position of the valve, collects data from frequency meters, and regulates the gas flow rate via voltage supplied to the pump’s engine. The microcontroller is connected to a computer using „QCM enos” v6 software (Gdansk University of Technology, Gdańsk, Poland), which allows adjusting of the operation parameters, data recording, and graphical presentation of the frequency on a plot. The measurement procedure was unified according to the protocol presented before [60]. To confirm the negative responses, the response of bare QCM electrodes was monitored prior to all measurements (Figure 4). All measurements were made with the bioelectronic nose system presented in Figure 8.

## 5. Conclusions

This article concerns an increase in biosensors’ sensitivity to aldehydes via the incorporation of different linker types between the peptide receptor and C-terminal cysteine residue binding QCM gold surface. This is the first study where the effect of linker incorporation into the peptide sequence was used to increase the binding affinity of gas molecules. Suitable optimization of the length and sequence of a linker chain allows improvement of sensitivity to the long-chain aldehydes present in the gas phase. A biosensor with a GSGSGS linker incorporated into the OBPP4 peptide exhibited LOD comparable to sensors reported in the literature. It is noteworthy that its usage does not require long exposure time to the gas phase and pre-heating stages. Essentially, the presented strategy of stabilization of the peptide’s receptor structure via the introduction of the linker is universal and should be compatible with other biosensor platforms. Theoretically, it is impossible to take into account all interactions occurring during ligands docking; there are indeed discrepancies between in silico docking and experimental results. A key to the actual tuning of receptor vs. ligand affinity is experimental studies with gas mixtures. Summarizing, the addition of the linker to the peptide’s structure provides more effective docking of volatile ligands; glycine-serine linkers provide the highest increase in affinity, which yielded an increase in biosensor’s sensitivity by ca. 15% with respect to the original peptide. The highest sensitivity values were also obtained for the GSGSGS linker: 0.3312, 0.4281, and 0.4676 Hz/ppm for pentanal, octanal, and nonanal, respectively. Generally, the dependence between the rigidity of a linker and the number of amino acid residues is much more pronounced than variations in a sequence. It was found that longer linkers had a better influence on docking effectiveness; the selection of the linkers with suitable length and sequence can constitute an additional aspect during the construction of more effective peptide-based biosensors for gas substances. In further studies, hydrocarbon stapling can be considered to stabilize peptide secondary structure [86]. It is expected that some conformational constraints can improve affinity to aldehydes. Moreover, peptides obtained by Solid Phases of Peptide Synthesis are typically trifluoroacetate salts (TFA). TFA anions are counter-ions for amine groups of lysine residues and N-terminus (TFA^−^⋯^+^H_3_N^−^). Moreover, amine groups are believed to interact with aldehydes (Schiff base), and presumably occurrence of TFA^−^ can affect these interactions. Therefore, further studies should include different counter-ions to evaluate their effect on biosensors. Additionally, biosensors will be employed in the array to test their response to real samples of exhaled breath.

## Figures and Tables

**Figure 1 ijms-24-10610-f001:**
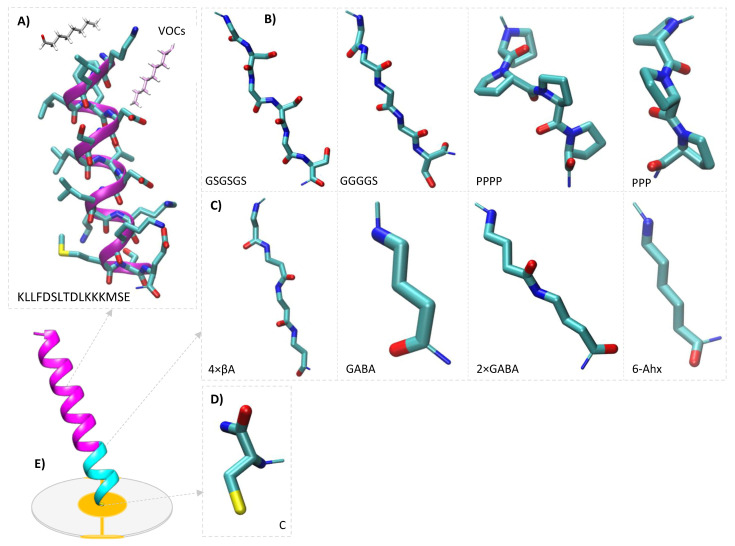
Schematic presentation of quartz crystal microbalance-based biosensor structure (**A**), composed of primary transducer receptor part (**B**), different types of linkers (**C**), (**D**) aimed at stabilization of the receptor part and cysteine residue binding the peptide to the gold electrode of secondary transducer (**E**).

**Figure 2 ijms-24-10610-f002:**
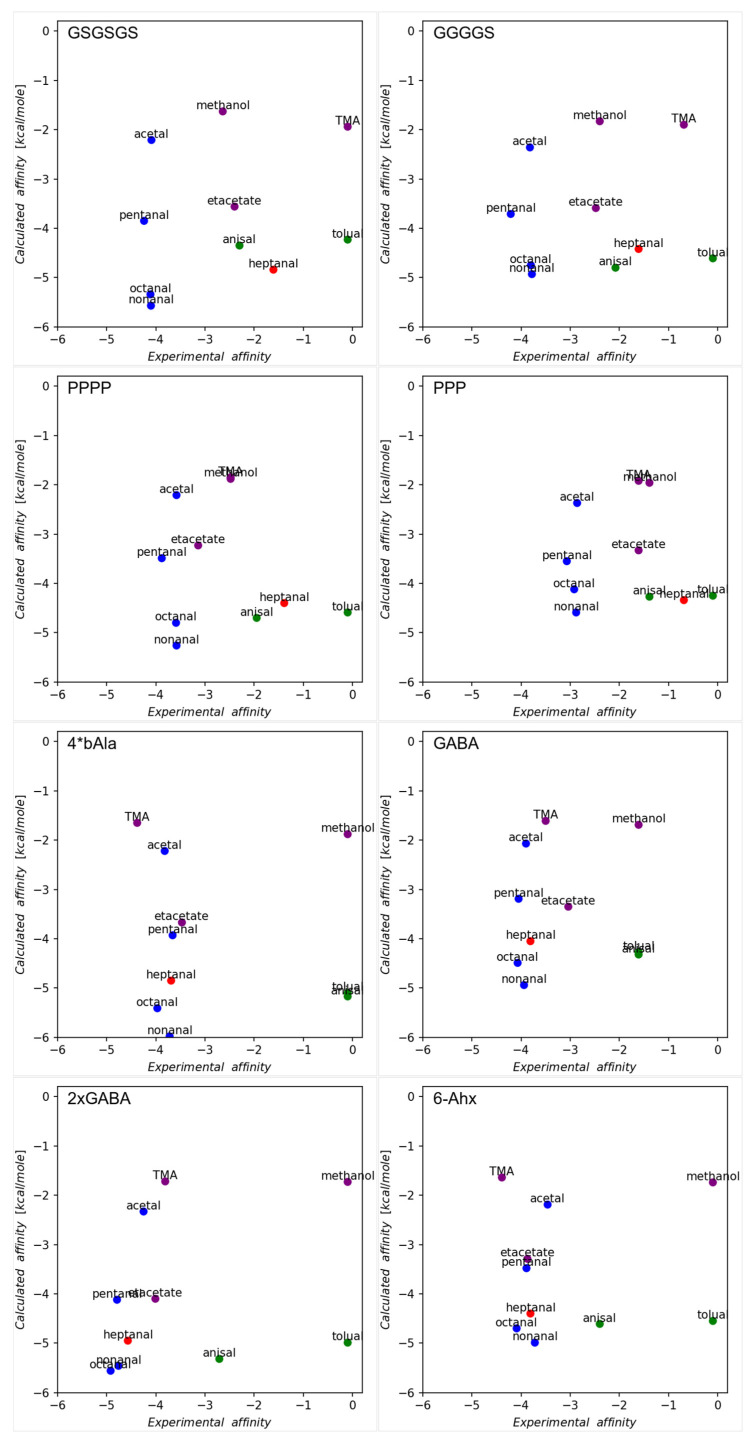
Comparison of the in silico docking predictions with the results of experiments. The calculated affinities are presented as reported by Autodock in kcal/mole, whereas the experimental affinity values are calculated as –ln(ΔF) (where ΔF stands for the averaged difference in frequency between the value recorded as the baseline (F_0_) prior to adsorption and the output frequency defined as F_R_) so that both sets of measurements are in comparable ranges for better comparison. Heptanal, aromatic aldehydes, and non-aldehyde VOCs are represented by the colors red, green, and purple, respectively. The remaining aldehyde VOCs are depicted as blue dots.

**Figure 3 ijms-24-10610-f003:**
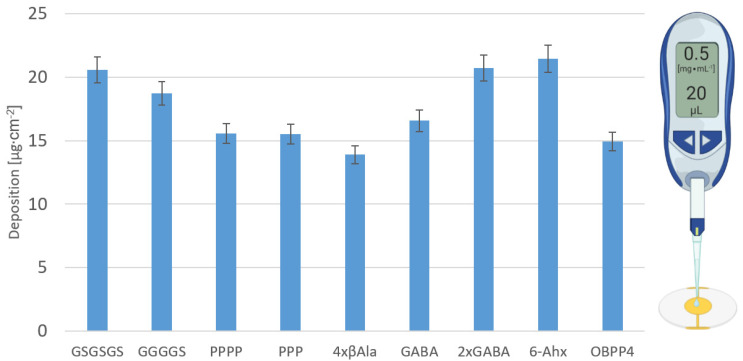
Peptide packing density on secondary transducers of QCM type.

**Figure 4 ijms-24-10610-f004:**
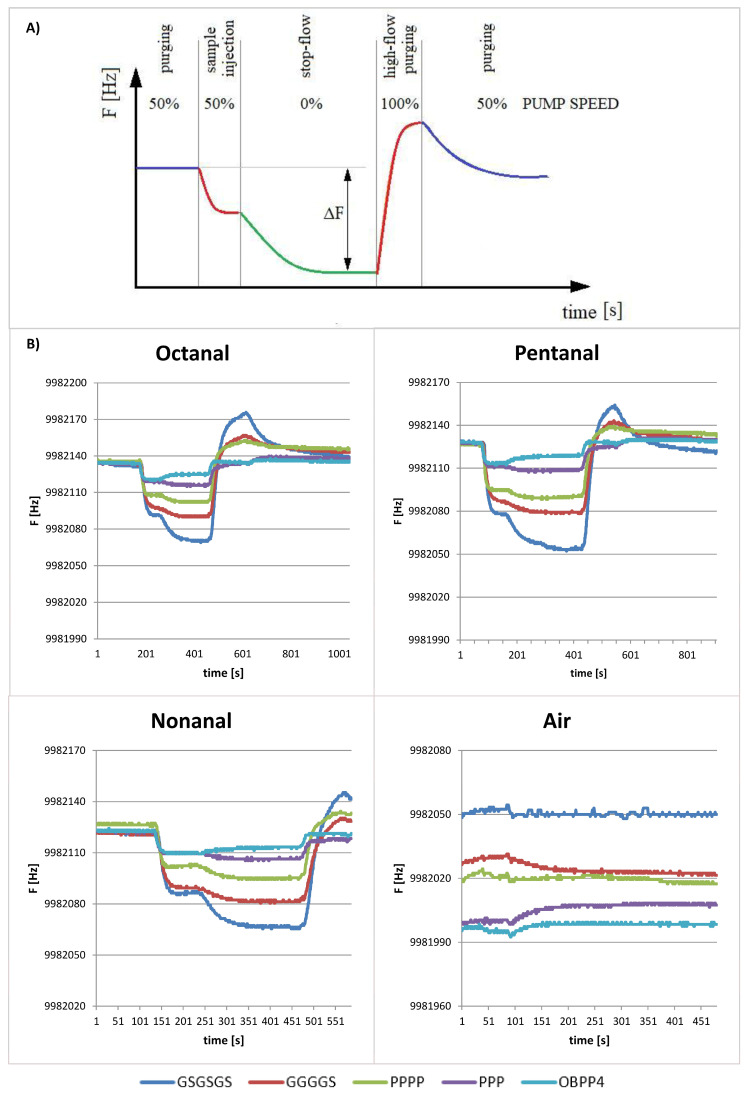
(**A**) The individual stages of the gas sample analysis and resonant frequency responses of peptide (OBPP4) Baseline of the sensors were established by flushing dry air, then sensors were exposed to a specific concentration of VOCs. After the introduction of a gas phase, the sensor frequency was reduced until the steady state was reached due to the maximum adsorption of gas molecules. Finally, the return to the initial sensor baseline was achieved by replacing VOCs with zero air. (**B**) Responses of peptides-based biosensors with linkers from group I and OBPP4 alone to long-chain aldehydes (octanal—31 ppm, pentanal—45 ppm, nonanal—28 ppm) and pure air with signal drift.

**Figure 5 ijms-24-10610-f005:**
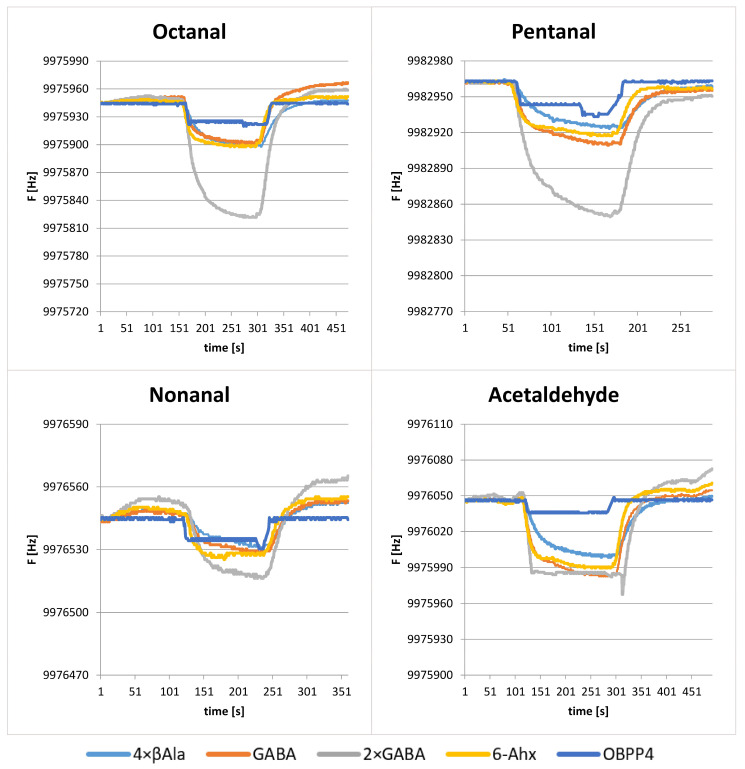
Resonant frequency responses of peptide (OBPP4) and peptides with linkers from group II to long-chain aldehydes (octanal—31 ppm, pentanal—45 ppm, nonanal—28 ppm, and acetaldehyde—86 ppm).

**Figure 6 ijms-24-10610-f006:**
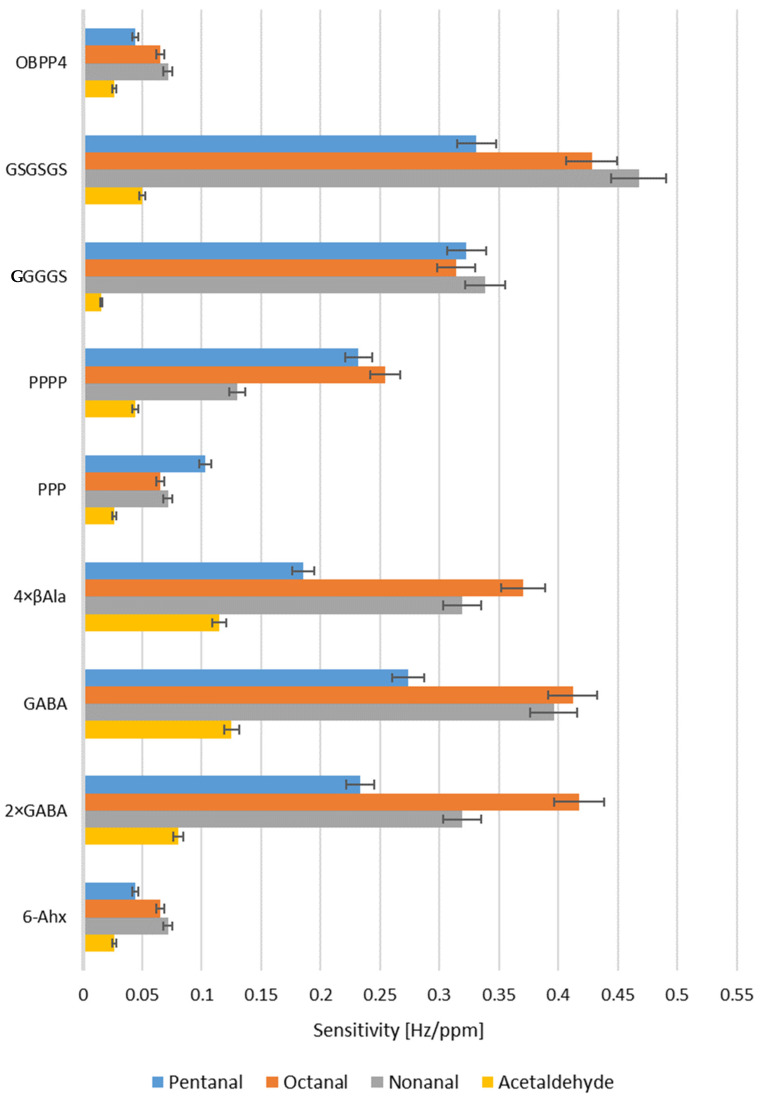
Comparison of biosensors’ sensitivity to aldehydes.

**Figure 7 ijms-24-10610-f007:**
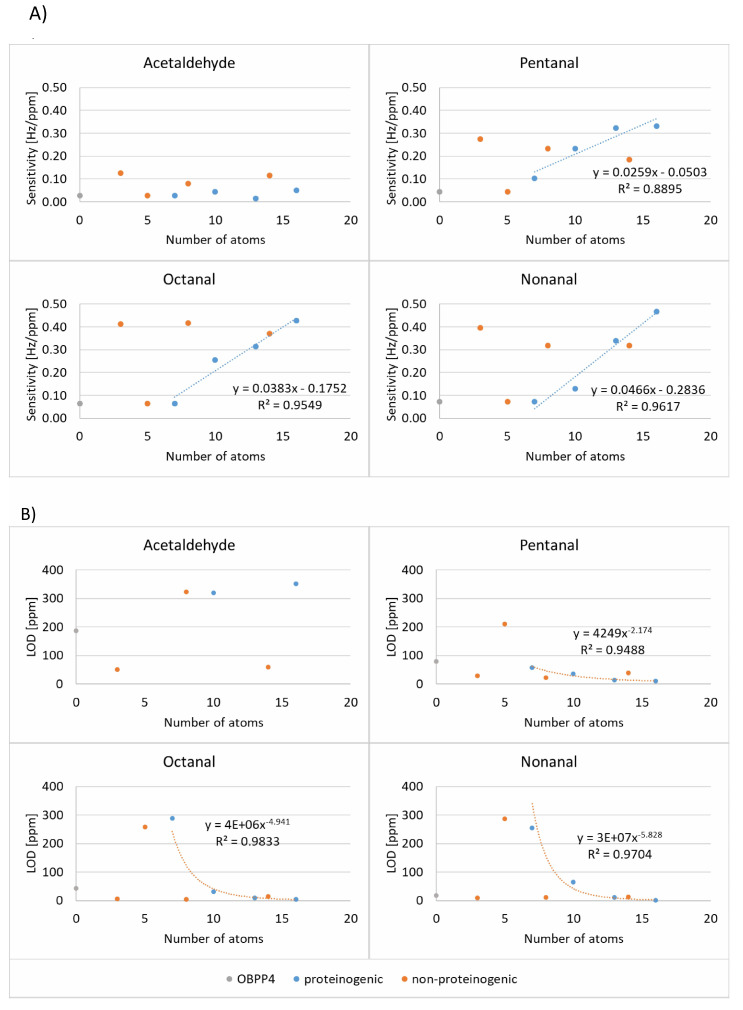
(**A**) The number of linker atoms vs. sensitivity to acetaldehyde, pentanal, octanal, and nonanal. (**B**) The number of linker atoms vs. LOD of acetaldehyde, pentanal, octanal, and nonanal.

**Figure 8 ijms-24-10610-f008:**
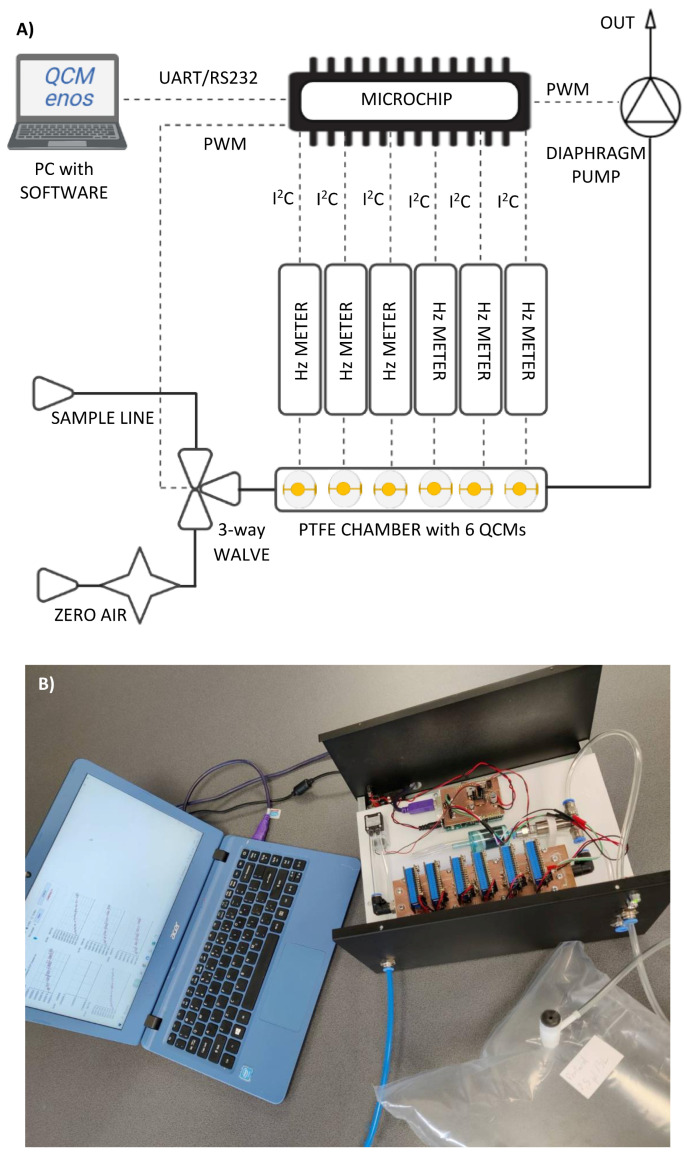
Measurement setup: (**A**) schematic representation of a bioelectronic nose for the detection of gas samples, (**B**) a real view of the instrumentation.

**Table 1 ijms-24-10610-t001:** List of linkers. The calculated length of linkers is based on the results of the molecular dynamics simulations. The reported values are the average distances between the amino group and terminal carbonyl carbon of the respective linker in its most abundant conformation.

Linker	Number of AA Residues	Number of Atoms Between Amino Group and Terminal Carbonyl Carbon Atom	Calculated Length [nm]
Group I—proteinogenic			
GSGSGS	6	16	0.94
GGGGS	5	13	1.40
PPPP	4	10	1.14
PPP	3	7	0.84
Group II—non-proteinogenic			
4 × βAla	4	14	0.95
GABA	1	3	0.56
2 × GABA	2	8	0.90
6-Ahx	1	5	0.66

**Table 2 ijms-24-10610-t002:** Comparison of sensitivity and limit of detection (LOD) for biosensors towards aldehydes.

Peptide	Sensitivity [Hz/ppm] LOD [ppm]			
	Pentanal	Octanal	Nonanal	Acetaldehyde
OBPP4	0.0442	0.0649	0.0715	0.0265
	79	43	18	187
GSGSGS	0.3312	0.4281	0.4676	0.0503
	11	5	2	352
GGGGS	0.3228	0.3141	0.3382	0.0155
	13	10	11	-
PPPP	0.2319	0.2544	0.1298	0.0442
	35	32	65	320
PPP	0.1028	0.0649	0.0715	0.0265
	58	289	255	-
4 × βAla	0.1853	0.3702	0.3188	0.1149
	39	15	13	59
GABA	0.2738	0.4123	0.3961	0.1250
	29	7	9	50
2 × GABA	0.2331	0.4176	0.3188	0.0802
	22	5	11	322
6-Ahx	0.0442	0.0649	0.0715	0.0265
	211	259	287	-

**Table 3 ijms-24-10610-t003:** Comparison of sensors and biosensors for the detection of nonanal in the gas phase.

Recognition	LOD [ppm]	Ref.
Chemoresistive, SnO_2_ nanosheets; SnO_2_ nanosheets and nanoparticles	0.1; 0.05	[74,75]
Chemoresistive, Pt-, Pd-, and Au-loaded SnO_2_ thick films	0.05	[76]
Chemoresistive, functionalized rGO	25	[80]
Chemoresistive, Polyetherimide/carbon black	1	[73]
Chemoresistive, MIP-AuNPs	4.5	[78]
Optical, vanillin	0.125	[77]
Piezoelectric, molecularly imprinted sol-gel	Several ppm	[81]
Piezoelectric, peptide	14	[60]
Piezoelectric, peptide	2	This study

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
