# Peer review of "Evaluation of Linkers’ Influence on Peptide-Based Piezoelectric Biosensors’ Sensitivity to Aldehydes in the Gas Phase"

_ijms, 2023, doi:10.3390/ijms241310610_

Round 1

Reviewer 1 Report

In this study, authors discuss

Evaluation of linkers influence on peptide-based piezoelectric 2

biosensors’ sensitivity to aldehydes in gas phase

The paper showed good results in relative major and I think need some correction before final acceptance. I suggest to authors that more attention to questions and comments

-        There are not any evidence about novelty of work and also not any discussion about this issue

-        Abstract is one of the important section in paper and need improve with more data

-        Authors must be compare this work and results with some other papers in this major

-        Conclusion need improve with some more data

-        Discussion of work must be improve and confirm with some data

-        Purity materials must be clear

-        Electrochemical methods have gained increasing attention in recent years as viable

-        alternatives to traditional analytical techniques due to their inherent advantages, such

-        High number of deaths caused by multifactorial diseases (cancers, respiratory system 42

-        diseases, cardiovascular disorders, infections, etc.) results mainly from late diagnosis, 43

-        which effectively limits treatment and significantly increases the cost of medical car must eb confirm with

-        https://doi.org/10.1016/j.chemosphere.2023.138815

Paper must be improve by a native person

Author Response

Responses in the attached file

Reviewer 2 Report

The manuscript “Evaluation of linkers influence on peptide-based piezoelectric biosensors’ sensitivity to aldehydes in gas phase” by Wasilewski et al. reports on the influence of the linker between the cysteine linking to a gold surface and an already-known sensing peptide for aldehydes on the proficiency of this sensing peptide.

While the results on precisely this topic could be of some interest (even if the sensitivity increase is marginal, just 15 %, as stated in line 492), the manuscript can be improved in its organization and in the discussion of the results; in particular, the explanations for the different effects of the different linkers are not solid enough.

1) There are many parameters which are not defined, possibly redefined with respect to their standard use, or with no clear (or with different than standard) experimental determination procedure. For example: “affinity” (Figure 1 and its description); most of the terms in lines 283 (and I would like to see some results for the systems considered in this manuscript), and in particular the LOD: it is not clear if the obtained values can be promptly compared with ones reported in the literature (determined using more standard and precise protocols), also because nowhere is stated how “signal” and “noise” were precisely determined/defined (and I would like to see calibration curves with clearly-defined error bars). Moreover, I am not sure if “selectivity to aldehydes” (see e.g. lines 449-450) measurements are reported in the paper, also because all reported results seem to regard some type of aldehyde.

2) In the manuscript, there are discussions about “stapled” peptides or secondary structures (e.g., lines 128-136, lines 502-504 and ref. 81). This is misleading or useless, since the studied systems are just normal linear peptides (as seen also from the graphical abstract).

4) Some experiment descriptions should be revised: e.g., in Figure 3 panels, the times of insertion of the tested gas and of the start of flushing with air should be indicated. Moreover, what are the reasons for the additional abrupt changes in the curves' derivatives? Can you clarify where is reported “the response of a bare QCM electrodes” stated in lines 475-476?

5) The results of computational simulations are poor and/or poorly described and/or not exploited enough. First, the results presented in Fig. 1 denote a very poor comparison between in-silico and experimental results (“properly defined” affinities experimentally within a decade could span up to 5 orders of magnitudes in simulation, and almost vice versa), in contrast with what is written in lines 300-301 (I don’t see a real “confirmation” in the presented data) and in line 488 (discrepancies are not “slight”, as instead stated there). I would like to see some results at least of the final structures in the computational simulations, at least in a supplementary figure. Better and better-exploited simulation results could have really helped in understanding why different linkers have different influences on the binding affinity of the active part of the peptide. This main result is not really explained on a solid basis within the manuscript, as also shown by all the “probably” and “it is supposed” in the discussion (e.g. lines 317, 320).

6) Is water considered in the simulations or at least in some of them? Indeed, and in any case, the native protein works in a “wet” environment. Is there any water trapped on the coating of the sensor? What are the used gas compositions? If they are dry, how do you think the results could change with real breath, usually containing a lot of water vapour? In any case, the gas mixture compositions cited in line 448 and the results of their verification described in lines 447-449 should be reported somewhere (in a supplementary figure?).

7) Some sentences in the discussion and in the materials and methods sections are better suited for the introduction (or should be moved from the materials and methods to the discussion): e.g., lines 349-356 and Table 3; lines 378-399 and Figure 8, which should be presented almost at the beginning and better explained (moreover, why are there different thicknesses for the bonds?); maybe lines 467-468 (and/or please report somewhere some time constants, and explain better if these depend on the degree of absorption of ligands, and in case why). In order to shorten the introduction, maybe some parts could be moved from there to the discussion or conclusion (especially if they can be better understood knowing some results).

Finally, the Authors should check carefully the English (e.g., use of articles, punctuation, clarity, some typos), and consider more specific points reported below.

In the abstract, lines 27-28: the linker is attached between the cysteine (and not its side chain) and the receptor-peptide.

Please, revise the first paragraph of the introduction: it is difficult to read, with some repetition and some too short sentences.

Line 35: what are the “variables in a sequence”?

Line 36-37: please rephrase.

Line 66: a “transducer” is not mentioned before; you may introduce it earlier, when talking about the sensor.

Line 108-109: that is not necessarily true.

Lines 135-136: check the sentence. Seems to be a fragment.

Line 143: which QCM data?

Lines 171-172: not all non-proteinogenic amino acids are more flexible, in general; the ones you chose are, because there are more carbon atoms than the alpha one between the amine and the carboxyl moieties.

Figure 1: not all compounds' names are readable; moreover, not all of the reported ones seem to be listed in the material section, or the abbreviations should be defined.

Line 234: it is not clear what the gas-nonanal reference is or how it was considered in the tests.

Line 287: check “corresponded”.

Line 291: correct figure numbers.

Line 336: I believe “exponential functions” should be substituted with “power laws”.

Line 345: wrong unit for “sensitivity”.

Line 470: I do not think the paragraph refers only to nonanal.

Line 496-497: I cannot understand what is exactly this threshold and why it exists; please rephrase, as asked for lines 36-37 in the abstract.

Lines 497-498: On which basis “it is expected that this phenomenon can influence on efficiency of catalytic sites”? And, which phenomenon, exactly?

The Authors should check carefully the English (e.g., use of articles, punctuation, clarity, and some typos).

Author Response

Responses in the attached file. 

Reviewer 3 Report

The manuscript “Evaluation of linkers influence on peptide-based piezoelectric biosensors’ sensitivity to aldehydes in gas phase” by Wasilewski et al. reports an increase of biosensors sensitivity via incorporation of different linker types between the receptor and docking parts. Although, some preliminary results are included, this work seems premature. Therefore, I would suggest authors may take at least a major revision before resubmission. Here are the comments and suggestions:

1.     Please add standard deviations to the results throughout this manuscript.

2.     In Fig. 1, the correlation seems not high enough. Did authors ever correct the coverage of these OBPP4 with different linkers with Fig. 2?

3.     Results shown in Figs. 3 and 4 are quite strange, and without any explanation on double plateau and overshooting after desorption.

4.     Calibration curves of every measurement should be added to the calculation of sensitivity.  

5.     In Table 3, the performance of this work seems not better than that in literature.

Author Response

Responses in the attached file. 

Reviewer 4 Report

In general, the authors did an excellent job on this research, and the manuscript is written logically, with the overall claim supported by the results. The article's structure is simple, with a broad discussion of the importance of piezoelectric biosensors and their extensive applications in gas-phase aldehydes detection.

The biosensor construction and aldehyde detection are accompanied by in-silico studies for validation, where the authors highlight the differences between the in-silico docking and experimental results.

I don't have concerns about the scientific part, and the results of the current study are fascinating. Nevertheless, it would add value to the present manuscript if the authors could discuss these biosensors regarding their reusability.

Author Response

Responses in the attached file. 

Round 2

Reviewer 2 Report

The manuscript improved considerably. There still are however some points that need clarifications, and some tweaks required in the text. In order to speed up the revision, I am attaching the pdf file with my annotations (disclaimer: some minor corrections, that I do as a habit while reading, could be not the best ones… Feel free to change them!).

I am especially concerned by:

1) the fact that not all the discussed quantities are properly defined (not even in [60]), and these definitions are necessary especially in cases where some terms are flawed; e.g., in the literature “sensitivity” is used both as “the minimum appreciable change in the measured quantity” (more similar to the LOD), and as “responsivity” (change in the actually “read” parameter, ΔF in this case, upon change in the measured quantity; related to the calibration curve);

2) the misleading data presentation in Figure 2 (even if the situation has improved; see comment in the pdf file, and please add units at least on the y-axis label);

3) the fact that not all data reported in the text are in agreement with the ones reported in tables and graphs (see comments in the pdf file).

Please, check carefully figures numbering and citations.

And I am also concerned by the fact that not all data/results are available (in supplementary files or in repositories?); could this be a problem for the “open data” policy now mostly required in scientific reporting?

Just minor tweaks are required.

Author Response

Dear Editors, 

Dear Reviewers, 

We would like to express gratitude for professional handling of our manuscript submitted to International Journal of Molecular Sciences entitled Evaluation of linkers influence on peptide-based piezoelectric biosensors’ sensitivity to aldehydes in the gas phase. We are grateful to the Editor and Reviewers for analysis and valuable remarks. The reviews outline interesting substantive issues and contain valuable observations.  

Below we enclose our responses to the Reviewers’ remarks. We hope they are comprehensive and satisfactory for the Reviewers as well as for the Editor, making our manuscript suitable for publication. Additions and subtractions from the body text of the manuscript were marked up in the current version of the manuscript using “Track Changes” function and highlighted.  

  • Reviewer 2

We want to extend our appreciation for taking the time and effort necessary to provide those comments and recommendations. We have revised paper in the light of those useful suggestions. We hope our revision improve the paper to a satisfactory level. 

The manuscript improved considerably. There still are however some points that need clarifications, and some tweaks required in the text. In order to speed up the revision, I am attaching the pdf file with my annotations (disclaimer: some minor corrections, that I do as a habit while reading, could be not the best ones… Feel free to change them!). 

I am especially concerned by: 

1) the fact that not all the discussed quantities are properly defined (not even in [60]), and these definitions are necessary especially in cases where some terms are flawed; e.g., in the literature “sensitivity” is used both as “the minimum appreciable change in the measured quantity” (more similar to the LOD), and as “responsivity” (change in the actually “read” parameter, ΔF in this case, upon change in the measured quantity; related to the calibration curve);
Response: According to the reviewers comment, sensitivity and LOD values were comprehensively described in the final version of the manuscript to be more accurate.

2) the misleading data presentation in Figure 2 (even if the situation has improved; see comment in the pdf file, and please add units at least on the y-axis label); 

Using different colors for aldehydes and other compounds could help in interpreting the graphs. 
Response: Both Figure 2 and its caption have been revised to incorporate the reviewer's comments 

3) the fact that not all data reported in the text are in agreement with the ones reported in tables and graphs (see comments in the pdf file). 
Response: Thank You for pointing out small mistakes and typos, all of them were corrected in the final version of the manuscript.  

Line 203: repetition was deleted. 

Line 216: check 
Response: Thank you for your comment. For the sake of clarity, the sentence was rephrased. 

Line 300: It is not clear if you are talking about the true experimental or the computational results.
Response: The text has been rephrased to clearly indicate that it refers to experimental results. 

Line 317: Not so clear; are you talking about the formation of covalent bonds? Or hydrogen exchange?  

Response: Line 358, Figure 2: So, they are two completely different quantities? Why should they be equal? Unless you demonstrate that -ln(ΔF) is (proportional to) the affinity in kcal/mole, there could be any functional dependence between the two... Please remove the diagonal line, it lets the reader think that the quantities in the two axes should be equal. And please anticipate in the caption what ΔF is (at the first reading, I thought it was a change in free energy...), and for which VOC concentration it was measured. 

Line 379: Not defined, even if understandable; jargon?

Response: Zero air parameters were added in Materials and methods section. Moreover, zero air term is a routinely used term – 10.1021/acs.analchem.8b04297.  In experiments zero air was obtained using GPZ-3B zero air generator (LAT Katowice, Poland). The air quality from this generator meets the following standards: EN 12619,EN 14211;EN 14212;EN 14625;EN 14626; EN 14662-3. 

Line 447: Determined at which concentration of the compound? not clear.  

Response: Thank You for the comment. We have added the formula for standard deviation of the response calculations in the text. 

Also, not even in [60] is reported the "calibration curve" used for the fit, whose R^2 you report.  

Response: In our studies we use linear regression for calibration. The determination coefficient is calculated for this regression. We added the information in the manuscript. 

Revise; maybe specify that the definition of some parameters is reported in [60], even if I could not find all the definitions even there... 

Response: Thank you for the correction. We have added this information in the manuscript. 

Line 524: You stated above that this does not form an helix. 
Response: Thank you for your comment. Short paragraph was added to maintain the logic of the discussion. 

Line 729: The increase seems to be much higher, not just by a factor 1.15; did you maybe mean that some LOD became 15% of the original? See table 2, check, and specify better the considered parameter and the ligand.  

Response: Thank You for the comment. Changes in sensitivity and LOD were corrected in the final version.  

Please, check carefully figures numbering and citations. 

Response: Thank You for the suggestion. All figures' numbers and citations were carefully checked.  

And I am also concerned by the fact that not all data/results are available (in supplementary files or in repositories?); could this be a problem for the “open data” policy now mostly required in scientific reporting? 

Response: Thank You for the advice. Crucial data regarding sensors responses and molecular modelling were included in the manuscript and supplementary files. 

It is our sincere hope that these changes improved the clarity and readability of the manuscript and made it more useful for potential readers of International Journal of Molecular Sciences.  

Kind regards, 

The Authors